# Mind the gap: Data availability, accessibility, transparency, and credibility during the COVID-19 pandemic, an international comparative appraisal

Arianna Rotulo[1], Elias Kondilis[2], Thaint Thwe[3,4], Sanju Gautam[5], Özgün Torcu[6], Maira Vera-Montoya[7], Sharika Marjan[8], Md. Ismail Gazi[9], Alifa Syamantha Putri[10], Rubyath Binte Hasan[11], Fabia Hannan Mone[12,13,14], Kenya Rodríguez-Castillo[15], Arifa Tabassum[16], Zoi Parcharidi[2], Beverly Sharma[17], Fahmida Islam[18], Babatunde Amoo[19], Lea Lemke[20], Valentina Gallo[1]*

1 Department of Sustainable Health, Campus Fryslân, University of Groningen, Leeuwarden, The Netherlands, 2 School of Medicine, Aristoteles University, Thessaloniki, Greece, 3 Department of Health Sciences, University Medical Centre Groningen, University of Groningen, Groningen, The Netherlands, 4 Graduate School of Medical Sciences, University of Groningen, Groningen, the Netherlands, 5 Department of Public Health, Faculty of Health Science, University of Southern Denmark, Odense, Denmark, 6 Faculty of Medicine, Ege University, Izmir, Türkiye, 7 Epidemiology Unit, Universidad del Cauca, Popayán, Colombia, 8 Department of Global Health, University of Bergen, Bergen, Norway, 9 Department of Public Health, Daffodil International University, Dhaka, Bangladesh, 10 Research Center for Public Health and Nutrition, National Research and Innovation Agency, Cibinong, Indonesia, 11 Chittagong Veterinary and Animal Sciences University, Chittagong, Bangladesh, 12 Department of Paediatrics, Anwer Khan Modern Medical College Hospital, Dhaka, Bangladesh, 13 Department of Public Health, Independent University, Dhaka, Bangladesh, 14 Institute of Social Welfare & Research, University of Dhaka, Dhaka, Bangladesh, 15 Unit Xochimilco, Metropolitan Autonomous University, Mexico City, Mexico, 16 Maternal and Child Health Division, International Centre for Diarrhoeal Disease Research, Bangladesh, Dhaka, Bangladesh, 17 Sustainalytics, Timisoara, Romania, 18 Department of Public Health, North South University, Dhaka, Bangladesh, 19 African Field Epidemiology Network, Abuja, Nigeria, 20 Bachelor degree in Global Responsibility and Leadership, Campus Fryslân, University of Groningen, Leeuwarden, The Netherlands

* v.gallo@rug.nl

**Data Availability Statement:** All data are in the manuscript and/or supporting information files.

## Abstract

Data transparency has played a key role in this pandemic. The aim of this paper is to map COVID-19 data availability and accessibility, and to rate their transparency and credibility in selected countries, by the source of information. This is used to identify knowledge gaps, and to analyse policy implications. The availability of a number of COVID-19 metrics (incidence, mortality, number of people tested, test positive rate, number of patients hospitalised, number of patients discharged, the proportion of population who received at least one vaccine, the proportion of population fully vaccinated) was ascertained from selected countries for the full population, and for few of stratification variables (age, sex, ethnicity, socio-economic status) and subgroups (residents in nursing homes, inmates, students, healthcare and social workers, and residents in refugee camps). Nine countries were included: Bangladesh, Indonesia, Iran, Nigeria, Turkey, Panama, Greece, the UK, and the Netherlands. All countries reported periodically most of COVID-19 metrics on the total population. Data were more frequently broken down by age, sex, and region than by ethnic group or socio-economic status. Data on COVID-19 is partially available for special groups. This exercise

**Funding:** The authors received no specific funding for this work.

**Competing interests:** The authors have declared that no competing interests exist.

highlighted the importance of a transparent and detailed reporting of COVID-19 related variables. The more data is publicly available the more transparency, accountability, and democratisation of the research process is enabled, allowing a sound evidence-based analysis of the consequences of health policies.

## Introduction

As the COVID-19 pandemic is raging across the globe, scientific evidence on transmissibility [1] and the effectiveness of mitigation strategies [2] is accumulating. When translated into health policy, however, this evidence has produced divergent scenarios [3, 4]. The efficacy of each public health policy could be indirectly evaluated though COVID-19 related data, which have been made publicly available by the majority of the national health authorities, coordinated by the World Health Organisation (WHO) [5], since the very early days of the pandemic [6].

Data transparency has played a key role in this pandemic, facilitating the cross-country comparison of local and national policies, and their evaluation [7]. Collective research efforts on data analysis and prediction modelling have fostered dialogue between the scientific community, public health authorities, and policy-makers. However, in many contexts, data availability and transparency have been suboptimal [8, 9], a factor that brought a number of negative repercussions in many sectors.

One key aspect of this pandemic that emerged quite early during the process of data sharing was the magnification effect that COVID-19 had on social inequalities, between and within countries [10]. This was detected thanks to the availability of stratified data in some places, but wider availability and increased granularity would permit an even more refined assessment of inequalities. In particular, the breakdown of data reporting by age, sex, region, and ethnic group would help identifying vulnerable groups, which in turn could inform public health strategies and health policies [10]. Further stratification, e.g. reporting cases and deaths by occupation, or in specific subgroups (e.g. students, health workers, etc.) would be instrumental to identify patterns of social and health inequalities, and to effectively manage the epidemic at different levels of governance [8] and to ensure political transparency and accountability. To our knowledge, no scientific paper before assessed data availability, accessibility, transparency and credibility internationally, and their related policy implications.

The aim of this paper is to map the availability and transparency of COVID-19 data in selected countries, by source of information, by a number of stratifying variables, and in specific risk groups and to rate their accessibility and credibility. This information is used to identify knowledge gaps, and to analyse policy implications.

## Methods

The Summer School in "Sustainable Health—Designing a new, better normal after COVID-19" took place remotely between the 5th and the 10th of July 2021 at Campus Fryslân, University of Groningen. The Summer School attracted a total of 21 students from 14 different countries, from the five continents. All students were postgraduates with a medical/health-related or social science background. During the week, the students were invited to identify COVID-19-related data available in their own countries (either their country of origin or of residence, whose language they were proficient in) and to map their different sources. Each student filled in a shared spreadsheet prepared in advance by three co-authors (AR, EK, and VG). As part of the exercise, students were also invited to rank both the overall accessibility and the credibility

of the information. During the last session of the summer school, students were divided into groups and asked present their findings to the whole group. The sampling of countries included in this study is therefore opportunistic; however this choice was aimed at privileging the importance of local knowledge of the sources of information but also of the public discourse around COVID-19, in each of the countries [11]. This includes the relative weight of pressure groups, the controversy around policies, and the infiltration of fake news in shaping the public opinion (for example from Anti-Vaxxers groups).

## Extraction of data

Data was extracted when possible in pairs or small groups in order to ensure double checking and quality control. Summer school supervisors (AR and EK) were available throughout the process to answer any questions or query the students might have. They also run *ad hoc* session to showcase how to extract data, how to classify them, and how to fill in the shared spreadsheet. Unfortunately, the language barrier did not always allow a proper double checking of data extraction. Thus, some imprecision is still possible.

## Availability and transparency

The extraction tables were designed to be filled in with information from each of the included countries. Information to be collected included the *availability* to the following periodically reported items: i) number of new COVID-19 cases (incidence); ii) number of COVID-19 death (mortality); iii) number of people tested for COVID-19; iv) COVID-19 positive rate (number of those testing positive out of the total number of people tested); v) number of patients hospitalised with COVID-19; vi) number of patients discharged after being hospitalised for COVID-19; vii) proportion of the population who received at least one vaccine; viii) proportion of population fully vaccinated (2/2 or 1/01 at the time, depending on the types of vaccine).

The availability of the information described above was collected, by country, for the full population, and by a number of stratification variables and categories/subgroups. The stratification variables included: 1) age; 2) sex; 3) subnational regions; 4) ethnic background; 5) socio-economic status. Official sources were defined as those government-related sources, such as the Office for National Statistics in the UK. Unofficial sources were defined as non-governmental organisations (NGOs), associations, or established special interest groups (i.e. charities working in prisons, or trade unions).

The overall data *transparency* was qualitatively evaluated according to the number of special categories/subgroups data was regularly reported for. These were: a) residents in nursing homes; b) inmates; c) students; d) healthcare and social workers; e) refugees or residents in refugee camps. In addition, the availability of information on the number and size of outbreaks in long-term care facilities, refugee camps, prisons, schools/universities, factories, and nosocomial institutions was also recorded by country.

## Accessibility and credibility of information

By the end of the exercise, a questionnaire was distributed to all participants asking about the accessibility of data in their researched country, and an overall evaluation of data quality and credibility in function of the sources. For *accessibility*, students were asked to rate how difficult it was to find the relevant data from 1 (very easy) to 5 (very difficult). For *credibility*, they were asked to rate how credible they thought data coming from official and unofficial sources were from 1 (not credible at all) to 5 (completely credible). This judgment is subjective, based on informal knowledge of the discourse around COVID-19 data availability in their countries,

and it has been largely discussed during tutorials. It is important for this exercise because it values the informal knowledge of the context, which would be impossible to judge by only accessing the websites remotely [11].

### Public involvement in the research

This study is a student-teacher collaboration during the online Summer School in Sustainable Health at the University of Groningen. The activity was methodologically led by tutors (AR, EK) and relied on the expertise and contextual knowledge of the students, who contributed to the debate about the importance of data accessibility with examples from their contexts, and who also took an active role in writing this paper.

## Results

A total of nine countries were included in the exercise, with the UK being split into England, Wales, Scotland, and Northern Ireland, resulting in a total of 12 individual country policies. Of these, four (Bangladesh, Indonesia, Iran, Nigeria) were classified as lower-middle income countries; two (Turkey, Panama) as upper-middle income countries; and three (Greece, Netherlands, United Kingdom) as high-income countries, according to the latest World Bank classification [12]. For each country, at least one person fluent in the official language of the country and with some public health/health system background, familiarised with the main data repositories and websites of public relevance and was responsible for data searching and extraction (S1 Table).

### Data availability and accessibility

Data availability for the included countries is shown in Fig 1. All countries regularly reported the total number of COVID-19 cases, mortality, testing, hospital admissions, and vaccination from official sources periodically, with few exceptions: in Nigeria the COVID-19 positive rate (positive tests to total tests), and the COVID-19 hospitalisation rates were not available, while the proportion of the population partially vaccinated was accessible through unofficial sources. In the Netherlands, the number of people discharged after being treated for COVID-19 was not available.

Data availability of COVID-19 data per stratification variables (age, sex, region, ethnicity, and socio-economic status) are reported in Fig 1. Overall, data were more frequently broken down by age, sex, and region than by ethnic group or socio-economic status. Variations were observed in terms of disaggregated data in the same income category. The only countries reporting an adequate break-down per stratification variables were the countries in UK. The Netherlands, Greece, and Turkey reported some break-down by age, sex, and region only for incidence, mortality, and hospitalisation data. However, the information did not always come from official sources. Bangladesh and Indonesia reported some break-down by age, sex, and region only for incidence and mortality data. Iran, Nigeria, and Panama reported little to no broken-down data on all COVID-19 Indicators.

Overall, discharge after COVID-19 resulted to be the category with the least data available by stratification variables. COVID-19 incidence, mortality, and hospitalisation were the variables that more often were presented according to different stratification variables categories. Stratification of data on vaccination was reported only in the four UK countries, and partially in Indonesia, the Netherlands, and Greece.

On average, *data accessibility* was considered difficult: on a scale from 1 (very easy) to 5 (very difficult), the mode was 4 (difficult), rated so by 5 participants (35.7%).

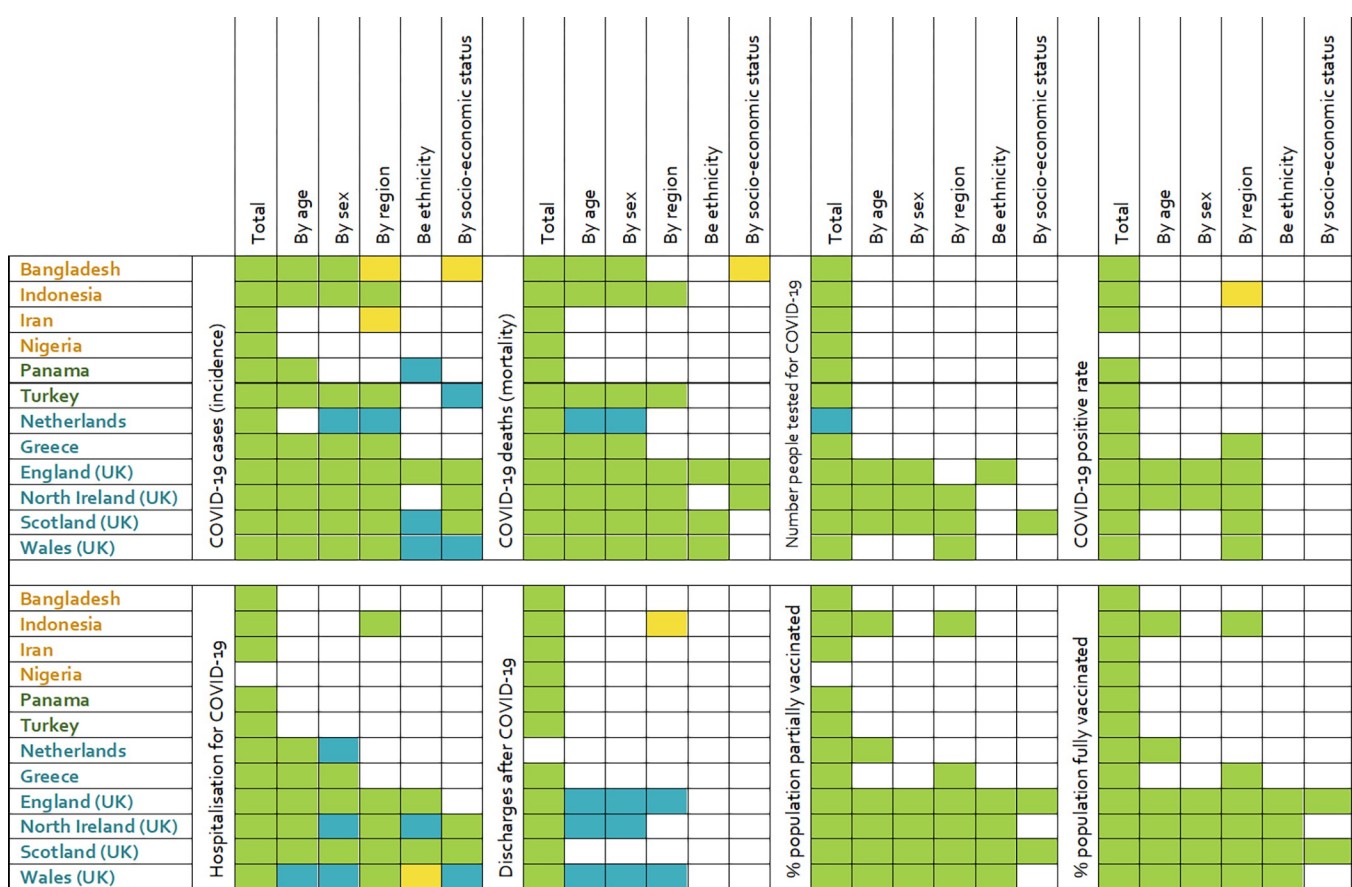

**Fig 1. Heatmap illustrating the availability of data on a number of COVID-19 variables in 12 selected countries (divided into lower middle-income in orange; upper middle-income in dark green; and high income in teal) in total, and by a number of stratifying variables.** Green: complete data from official source; yellow: incomplete data from official source; blue: data from unofficial source.

## Transparency of data reporting and credibility

Results of the analysis of data in special sub-groups (residents in nursing homes, inmates, students, healthcare & social workers, and refugees) are reported in Fig 2. Data mainly on COVID-19 incidence and mortality is partially available in a number of countries, while data for the other COVID-19 related variables are more scattered. The country which best reports data according to special categories is Scotland with official/unofficial sources covering most of the fields, particularly among students and health care workers (incidence, mortality, number of tests, positivity rate, vaccine). All UK countries, except for Northern Ireland reported data on vaccination among residents in nursing homes and healthcare and social workers, but data on COVID-19 incidence and mortality among healthcare and social workers is incomplete or coming from unofficial sources in England and Wales. COVID-19 among refugees was officially reported only by Bangladesh (incidence, mortality, and testing); in Greece data on incidence was partially complete, in England, Wales, and Scotland the data was reported by unofficial sources. Data on COVID-19 incidence among inmates was sporadically available from official sources only in Bangladesh, Indonesia, England, Northern Ireland, and Scotland; in Panama and the Netherlands the data was gathered from unofficial sources. Data on COVID-19 related mortality were also available in Indonesia, the Netherlands, England, Scotland, and Wales. Data on vaccination was available only in Indonesia from unofficial sources.

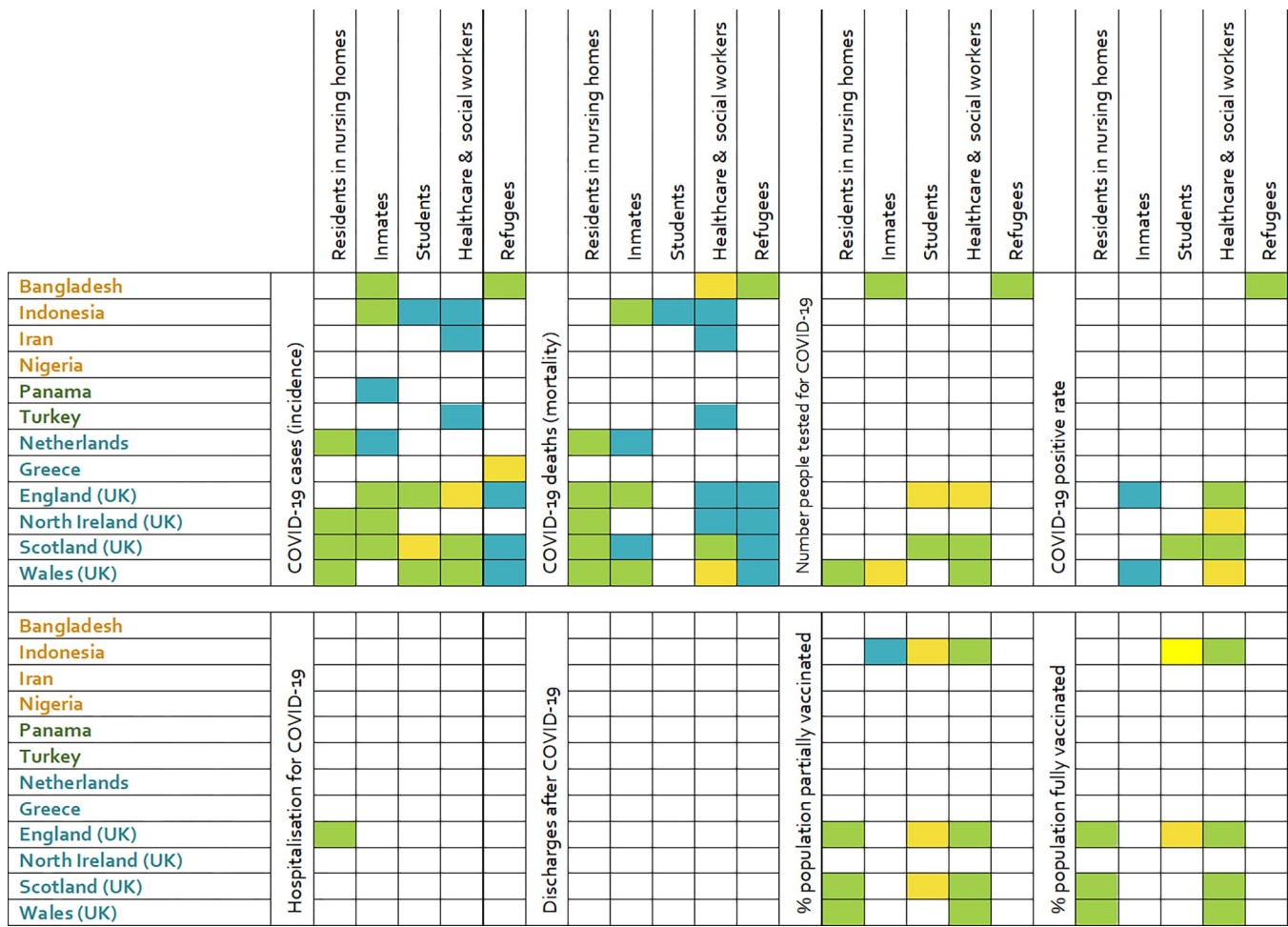

**Fig 2. Heatmap illustrating the availability of data in a number of COVID-19 variables in 12 selected countries (divided into lower middle-income in orange; upper middle-income in dark green; and high income in teal) in a number of population sub-groups.** Green: complete data from official source; yellow: incomplete data from official source; blue: data from unofficial source.

Interestingly, the majority of the authors who extracted the data rated the *credibility* of both official and unofficial sources as high: the mode being in both cases 4 (very credible) rated so by 5 participants (35.7%).

## Discussion

This paper reports a first attempt to appraise in a systematic way COVID-19 related data from a selected number of countries by type of data, stratification variables, and special sub-groups. It prompted a number considerations around the issue of data availability and transparency and the importance of these in pandemic management. Overall, the results suggest an unprecedented effort in collating and making epidemiological data publicly and widely available to the general public from trustworthy sources, despite the fact that such data were considered not always easy to find and access. Varying levels of available budget and infrastructures in high- and low-income countries have not generated significant differences in data availability and accessibility, at least for collated, not stratified data.

Access to stratified data is essential to uncover inequalities in COVID-19 morbidity [10, 13–16]. Among the included countries, the countries in the UK, and–to some extent–

Indonesia, had the most accessible data. Good geographical stratification in the UK countries identified in this paper, for example, was reported as refined data availability by Middle-Layer Super Output Areas (MSOA) in England, which allowed to explore the relative role of spatial inequalities and of structural factors in explaining the geographical distribution of COVID-19 mortality [15]. Similarly, in England and Wales availability of data by age and sex, identified in this paper, allowed to estimate the reduction of life expectancy at birth and lifespan inequalities in previous work [17]. Data broken down by ethnic group detected here as reported by the Office for National Statistics (ONS) in the UK [18], prompted a parliamentary investigation on why COVID-19 mortality rates were highest among people from Black, Asian, and Minority Ethnic (BAME) groups, with Black males 3.3 times more likely to die compared to their white counterparts [18, 19]. That investigation resulted in a report [20] suggesting that racism, discrimination, and social inequalities have contributed to the increased risks not only of infection but also of complications and death from COVID-19 among minority ethnic people [19]. Importantly, the report emphasised that longstanding inequalities affecting BAME communities in the UK were exacerbated by the conditions under which BAME people live [19]. Similar disparities based on ethnicity and migration status, although not included in the present analysis, were found in other countries such as Sweden [21]. The ethnic break-down for vaccine intake is also a crucial piece of information to identify groups whose uptake is suboptimal and to tailor appropriate public health campaigns, as shown by a study in US [22].

More generally, when sex-disaggregated data are available, observed inequalities within a country can be appraised in the light of the relative effect of biological factors [23, 24] and gender norms [25]. Differences between male and female rates of COVID-19 cases and deaths are larger in countries where women experience more discrimination within families and have less access to resources, education, and finance [26]. Sex-stratified data in the United States also suggested a different attitude toward vaccine intake between men and women [27]. Neglecting sex and gender differences in COVID-19 renders these gender/sex-specific challenges effects unobservable [27]. On the other hand, combining such information with data on ethnic background allows an intersectional approach to better understand the relative role of social and biological factors [28].

Data reporting broken down by geographical and demographic strata facilitates international comparison [29] and points out inequalities in varying country contexts [30–32]. In the context of vaccination uptake and availability, it can prompt reflections on vaccination equity and the success of the COVAX programme [33].

The transparency of reporting of COVID-19 incidence and mortality in special categories has contributed to a better understanding of the main mechanisms of transmission [34] and the role of inequalities [35], and occupational hazards [36], but has also increased transparency and accountability of health policy decisions. Having observed the very high number of COVID-19-related deaths in nursing homes in England, the UK High Court recently established that the decision–in spring 2020 –to discharge people from hospitals to care homes without mandatory isolation or testing was *irrational* and *unlawful* [37]. Data coming from special categories (i.e., prison inmates, people in detention centres and reception centres) can inform the issue of special guidelines for prevention in those contexts [38]. Nevertheless, COVID-19 data reporting for these categories remains specifically underreported and therefore understudied, as noted also previously [39].

The downside of increasing data availability and transparency is the potential violation of privacy protection. This might be particularly true when punctual data coming from individual institutions (e.g. prison, nursing home) are provided separately. However, central government offices can play a pivotal role here in modelling these data to a national representativeness level, while at the same time guaranteeing compliance with privacy

regulations. Previous research showed that the general population are willing to facilitate data sharing by for example actively using apps, i.e. the track and tracing one, if privacy and security protection are designed and implemented [40].

In this paper we show that the subjective evaluation of both official and unofficial sources is believed to be credible and overall of high quality. This aligns well with the fact that monitoring of the available COVID-19 data at the international level has been done by several institutions [5, 41] and initiatives [42, 43], not all of them from governmental official sources. Their work has been extensively used to analyse the rapidly evolving situation [6, 7], as well as to estimate international [44, 45] and national [46] interventions and policies, and their impact. On the other hand, the unavailability of timely and complete data can elicit misinformation and disinformation among the public, which eventually might hamper the overall health policy enforced [47]. The transparent, thorough, and complete report from national authorities has been the necessary first step to allow so.

At the time of writing, the world is living into its third year of COVID-19 pandemic, with an internationally shared sense of grief and fatigue, and uncertainty about the future. It is now more important than ever that the public maintains trust in the institutions [48] and follows government indications to test and receive vaccinations [49–51]. Trust in institutions is also likely to induce populations to share crucial information [52] thereby maintaining an effective surveillance system.

Among other things, trust can be enhanced by a transparent and detailed report of available data which increases the accountability of public health authorities [8]. Data transparency can also democratise the research effort in fighting the pandemic, ultimately promoting an evidence-based best practice less sensitive to vested interests and political agenda influences.

## Strengths and limitation

This study compares COVID-19 data availability, accessibility, transparency, and credibility in nine resource different and geographically distant countries. Importantly, it maps both official and unofficial sources of information and data access was performed by post-graduate public health/health system professionals who were familiar with the cultural context, the language, and the main reporting sources of each country. Despite their advantaged position as credible knowers [11], the challenges of navigating and finding official stratified data on multiple indicators remain daunting for students. This may have affected their search outcomes and performance. The inclusion of more countries would have increased the quality of the cross-sectional comparison. However, the present data aims at exemplifying the importance of detailed data reporting rather to provide a comprehensive picture.

## Conclusions

In conclusion, this exercise maps a varied combination of COVID-19 related data and their sources. Reported evidence highlighted the importance of a transparent and detailed reporting of COVID-19 related variables by public authorities. The more data is publicly available, the more the research process can benefit from transparency, accountability, and democratisation. This allows a sound evidence-based analysis of the consequences of different health policies. Through this mapping exercise, public health regulators can benchmark how well current information sharing policy is working in different parts of the world. The World Health Organisation can nudge public health authorities, leading the way in improving data sharing.

## Supporting information

**S1 Table. List of all official and unofficial sources of information used, by country.** (DOCX)

## Author Contributions

**Conceptualization:** Arianna Rotulo, Elias Kondilis, Valentina Gallo.

**Data curation:** Arianna Rotulo, Thaint Thwe, Sanju Gautam, Özgün Torcu, Maira Vera-Montoya, Sharika Marjan, Md. Ismail Gazi, Alifa Syamantha Putri, Rubyath Binte Hasan, Fabia Hannan Mone, Kenya Rodríguez-Castillo, Arifa Tabassum, Zoi Parcharidi, Beverly Sharma, Fahmida Islam, Babatunde Amoo, Lea Lemke.

**Formal analysis:** Thaint Thwe, Sanju Gautam, Özgün Torcu, Maira Vera-Montoya, Sharika Marjan, Md. Ismail Gazi, Alifa Syamantha Putri, Rubyath Binte Hasan, Fabia Hannan Mone, Kenya Rodríguez-Castillo, Arifa Tabassum, Zoi Parcharidi, Beverly Sharma, Fahmida Islam, Babatunde Amoo, Lea Lemke, Valentina Gallo.

**Investigation:** Elias Kondilis.

**Methodology:** Arianna Rotulo, Elias Kondilis, Valentina Gallo.

**Project administration:** Lea Lemke.

**Supervision:** Valentina Gallo.

**Visualization:** Valentina Gallo.

**Writing – original draft:** Arianna Rotulo, Valentina Gallo.

**Writing – review & editing:** Arianna Rotulo, Elias Kondilis, Thaint Thwe, Sanju Gautam, Özgün Torcu, Maira Vera-Montoya, Sharika Marjan, Md. Ismail Gazi, Alifa Syamantha Putri, Rubyath Binte Hasan, Fabia Hannan Mone, Kenya Rodríguez-Castillo, Arifa Tabassum, Zoi Parcharidi, Beverly Sharma, Fahmida Islam, Babatunde Amoo, Lea Lemke.

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
