## [Decision Letter · Decision Letter 0]

20 Feb 2023

PGPH-D-22-01479

Mind The Gap: Data availability, accessibility, transparency, and credibility during the COVID-19 pandemic, an international comparative appraisal

Dear Dr. Gallo,

Thank you for submitting your manuscript to PLOS Global Public Health. After careful consideration, we feel that it has merit but does not fully meet PLOS Global Public Health’s publication criteria as it currently stands. Therefore, we invite you to submit a revised version of the manuscript that addresses the points raised during the review process.

Two reviewers have evaluated your submission and both of them provided their comments below. Please provide your point-by-point responses to all the concerns.

We look forward to receiving your revised manuscript.

Kind regards,

Jianhong Zhou

Staff Editor

Journal Requirements:

1. Please amend your online Financial Disclosure statement. If you did not receive any funding for this study, please simply state: “The authors received no specific funding for this work.”

2. Please provide separate figure files in .tif or .eps format only and remove any figures embedded in your manuscript file. Please also ensure that all files are under our size limit of 10MB.

3. Please amend your Data Availability Statement and indicate where the data may be found

Additional Editor Comments (if provided):

Reviewers' comments:

Reviewer's Responses to Questions

**Comments to the Author**

1. Does this manuscript meet PLOS Global Public Health’s publication criteria? Is the manuscript technically sound, and do the data support the conclusions? The manuscript must describe methodologically and ethically rigorous research with conclusions that are appropriately drawn based on the data presented.

Reviewer #1: Partly

Reviewer #2: Partly

2. Has the statistical analysis been performed appropriately and rigorously?

Reviewer #1: N/A

Reviewer #2: I don't know

3. Have the authors made all data underlying the findings in their manuscript fully available (please refer to the Data Availability Statement at the start of the manuscript PDF file)?

Reviewer #1: Yes

Reviewer #2: No

4. Is the manuscript presented in an intelligible fashion and written in standard English?

Reviewer #1: Yes

Reviewer #2: Yes

5. Review Comments to the Author

Reviewer #1: 1. L149/150: Reconfirm data availability for partially vaccinated for Nigeria.

2. L155/156: 'the British ones' - Consistency in reporting on the UK. E.g. UK, Britain or England, Scotland and Wales. Or clarify preference at start.

3. L158: Information not always from official sources - This is debatable, not reliable and difficult to compare across countries. What are these other sources? Social media? Where data is not available from official sources can this be regarded as data for the country in context of this paper? This should be addressed. Suggestion is to include only officially available data.

4. L158-160: Nigeria reported on region (states), Number confirmed positive, number admitted, number discharged, number of deaths.

5. Consistency should be maintained in reporting on findings to make easy for reader to follow

6. L184/185: How was credibility of data of official and unofficial sources rated as high?? This is quite subjective thus should be reconsidered.

7. Improved clarity needed in the discussion section. E.g. It is not always clear what was a finding from this study vs reference to another study.

Reviewer #2: Thank you for giving me the opportunity to read this paper which talks about a timely topic.

Though the paper is written in a clear and structured way, I have a couple of comments:

• Method

- It would be helpful to clarify why the authors chose those nine countries. Was it just random because of who participated in the summer school? The authors should also be consistent in talking about 9 or 12 countries.

- I also wonder whether someone checked the data each student brought to the table. I think the methodology would have been stronger if each country had been researched by one person and checked/reviewed by another person to ensure accuracy. It does not appear to me that a second person reviewed the data (dual control principle). For example, how did you ensure that each student found all relevant data available in the respective country?

- Another big issue is that I haven’t seen a table that lists all links for each country where the authors have their data from. Maybe they submitted this, but I don’t see it in the submitted manuscript, and I cannot trust the figures without knowing the link/data sources from the different countries.

• Focus on disparities based on ethnicity /inequalities in COVID-19 morbidity

- I think the strength of the paper lies in the focus on the inequalities which can be read from the data. I think the author should mention this earlier on (already in the introduction and abstract) and make this issue the main focus of the paper.

• Data sharing versus privacy protection / surveillance

The authors demand more data sharing, but I believe they should also (at least briefly) discuss what this means for privacy protection/surveillance in addition to the trust issue they mentioned.

6. PLOS authors have the option to publish the peer review history of their article (what does this mean?). If published, this will include your full peer review and any attached files.

**Do you want your identity to be public for this peer review?** For information about this choice, including consent withdrawal, please see our Privacy Policy.

Reviewer #1: No

Reviewer #2: No

---

## [Decision Letter · Decision Letter 1]

29 Mar 2023

Mind The Gap: Data availability, accessibility, transparency, and credibility during the COVID-19 pandemic, an international comparative appraisal

PGPH-D-22-01479R1

Dear Dr. Gallo,

We are pleased to inform you that your manuscript 'Mind The Gap: Data availability, accessibility, transparency, and credibility during the COVID-19 pandemic, an international comparative appraisal' has been provisionally accepted for publication in PLOS Global Public Health.

Best regards,

Julia Robinson

Executive Editor

Reviewer Comments (if any, and for reference):

Reviewer's Responses to Questions

**Comments to the Author**

1. If the authors have adequately addressed your comments raised in a previous round of review and you feel that this manuscript is now acceptable for publication, you may indicate that here to bypass the “Comments to the Author” section, enter your conflict of interest statement in the “Confidential to Editor” section, and submit your "Accept" recommendation.

Reviewer #1: All comments have been addressed

Reviewer #2: All comments have been addressed

2. Does this manuscript meet PLOS Global Public Health’s publication criteria? Is the manuscript technically sound, and do the data support the conclusions? The manuscript must describe methodologically and ethically rigorous research with conclusions that are appropriately drawn based on the data presented.

Reviewer #1: Yes

Reviewer #2: (No Response)

3. Has the statistical analysis been performed appropriately and rigorously?

Reviewer #1: (No Response)

Reviewer #2: (No Response)

4. Have the authors made all data underlying the findings in their manuscript fully available (please refer to the Data Availability Statement at the start of the manuscript PDF file)?

Reviewer #1: (No Response)

Reviewer #2: Yes

5. Is the manuscript presented in an intelligible fashion and written in standard English?

Reviewer #1: (No Response)

Reviewer #2: Yes

6. Review Comments to the Author

Reviewer #1: (No Response)

Reviewer #2: All comment have been addressed. The new supplementary table improved the manuscript and transparency.

7. PLOS authors have the option to publish the peer review history of their article (what does this mean?). If published, this will include your full peer review and any attached files.

**Do you want your identity to be public for this peer review?** For information about this choice, including consent withdrawal, please see our Privacy Policy.

Reviewer #1: No

Reviewer #2: No
